# Carbohydrate Restriction in Type 1 Diabetes: A Realistic Therapy for Improved Glycaemic Control and Athletic Performance?

**DOI:** 10.3390/nu11051022

**Published:** 2019-05-07

**Authors:** Sam N. Scott, Lorraine Anderson, James P. Morton, Anton J. M. Wagenmakers, Michael C. Riddell

**Affiliations:** 1School of Kinesiology and Health Science, York University, Toronto, ON M3J 1P3, Canada; mriddell@yorku.ca; 2Independent Researcher, Toronto, ON M3J 1P3, Canada; lo.anderson16@gmail.com; 3Research Institute for Sport and Exercise Sciences, Liverpool John Moores University, Liverpool, L3 3AF, UK; j.p.morton@ljmu.ac.uk (J.P.M.); a.j.wagenmakers@ljmu.ac.uk (A.J.M.W.); 4LMC Diabetes & Endocrinology, 1929 Bayview Avenue, Toronto, ON M4G 3E8, Canada

**Keywords:** type 1 diabetes, low carbohydrate diet, carbohydrate, exercise training, insulin, glycaemia, train low, hypoglycaemia, very low carbohydrate diet, carbohydrate periodisation

## Abstract

Around 80% of individuals with Type 1 diabetes (T1D) in the United States do not achieve glycaemic targets and the prevalence of comorbidities suggests that novel therapeutic strategies, including lifestyle modification, are needed. Current nutrition guidelines suggest a flexible approach to carbohydrate intake matched with intensive insulin therapy. These guidelines are designed to facilitate greater freedom around nutritional choices but they may lead to higher caloric intakes and potentially unhealthy eating patterns that are contributing to the high prevalence of obesity and metabolic syndrome in people with T1D. Low carbohydrate diets (LCD; <130 g/day) may represent a means to improve glycaemic control and metabolic health in people with T1D. Regular recreational exercise or achieving a high level of athletic performance is important for many living with T1D. Research conducted on people without T1D suggests that training with reduced carbohydrate availability (often termed “train low”) enhances metabolic adaptation compared to training with normal or high carbohydrate availability. However, these “train low” practices have not been tested in athletes with T1D. This review aims to investigate the known pros and cons of LCDs as a potentially effective, achievable, and safe therapy to improve glycaemic control and metabolic health in people with T1D. Secondly, we discuss the potential for low, restricted, or periodised carbohydrate diets in athletes with T1D.

## 1. Introduction

Type 1 diabetes (T1D) is a chronic inflammatory autoimmune disease whereby the insulin producing β-cells in the pancreas are destroyed [1], resulting in lifelong dependence on exogenous insulin and management of blood glucose profiles through diet and exercise. Controlling blood glucose levels after a diagnosis of T1D is a constant challenge, with continuous lifetime risk of both severe hypoglycaemia and severe hyperglycaemia with ketoacidosis [2]. In the long-term, people with T1D are at increased risk of micro- and macrovascular complications due to lifetime exposure to imperfect glucose control [3,4]. The American Diabetes Association (ADA) treatment guidelines suggest a target HbA1c of <7.0% (53 mmol/mol) in adults with T1D (7.5% for youth with T1D) to reduce the risk of long-term diabetes-related complications [5]. Unfortunately, recent data from the T1D Exchange of 22,697 individuals with T1D in the United States demonstrated that only 17% of children and adolescents <18 years old, and only 21% of adults aged >30 years are achieving their HbA1c targets [2]. An alarming observation within this registry is that HbA1c levels are deteriorating, rather than improving, despite an increase in the use of recently developed technologies such as continuous glucose monitoring devices and insulin pumps [2]. The high proportion of individuals not achieving HbA1c targets with current therapies suggest that novel intervention strategies are urgently needed to reverse this deterioration in glycaemic control in this patient population. There is emerging evidence that dietary manipulation, and in particular low carbohydrate intake (<130 g per day), may be a relevant strategy for many people if weight loss and the prevention of both hypo- and hyperglycaemia is a concern [6].

For many people living with T1D, participating in physical activity on a recreational basis or achieving a high level of athletic performance is important. Achieving a balance between glycaemic control and nutritional intake to fuel performance is crucial for these individuals. Studies in people without T1D have shown the effectiveness of a periodic “train low, compete high” paradigm, whereby selected training sessions are deliberately undertaken with reduced carbohydrate availability to activate molecular pathways, in order to augment skeletal muscle adaptation (reviewed in [7,8,9]). Endurance training with reduced muscle glycogen stores (“training low”) has been shown to increase the expression of several genes encoding for mitochondrial enzymes, enzyme protein content and activity, and increase the rates of lipid oxidation during sub-maximal exercise when compared to training with normal or high glycogen stores [10,11,12]. For athletes with T1D, training with reduced carbohydrate availability may also represent a strategy that benefits glycaemic control in day-to-day life and training-induced increases in skeletal muscle oxidative capacity, which may have benefits both for endurance performance and body composition (i.e., reduced fat mass, increased lean mass). However, the long-term effects of training with reduced carbohydrate availability and consumption of low carbohydrate diets on skeletal muscle insulin sensitivity, metabolic health, training adaptation, exercise performance, and glycaemic control during and after exercise are yet to be investigated in people with T1D.

The aims of this review are as follows: (1) To investigate the known pros and cons of low carbohydrate diets (LCD) in people with T1D; (2) to discuss whether there is potential for low and/or restricted carbohydrate diets in recreationally active individuals with T1D; (3) to discuss carbohydrate restriction strategies for elite or recreational athletes with T1D as a means to improve training adaptation alongside improved glycaemic control. It is hoped that this paper will highlight areas for future research so that patients, healthcare providers, and sports coaches can make informed, evidence-based decisions about whether LCDs are suitable, while understanding the potential risks and limitations.

## 2. History of Carbohydrate Restriction in People with Type 1 Diabetes

In the pre-insulin era, carbohydrate and calorie restriction in the form of “starvation diets” were the most advanced treatment for T1D [13]. Frederick Allen and Elliot Joslin were both proponents of the “low carb” strategy, using prolonged fasting and carbohydrate restrictive diets, as this seemed to be the only way of delaying (although rarely preventing) death from ketoacidosis [14]. For more information, interested readers are referred to Hill and Eckman [15] for numerous historical reports of patients prescribed these starvation diets. With the discovery of insulin in 1922, the practice of dietary restriction remained, at least to some degree, and carbohydrate intake liberalisation was only very cautiously introduced. For the next 50 years, dietary advice for people living with T1D continued to focus on limited-to-moderate carbohydrate consumption with refined sugar restriction. Strict meal plans were calculated and “exchange systems” were developed, where food portions, and in particular carbohydrates, were carefully weighed and measured.

In the 1960s, commonly held beliefs around diet and T1D were challenged and concerns were raised about the risks of a low carbohydrate, high saturated fat diet in an atherogenically at-risk population [14]. Recommendations shifted to reduce fat intake (<35% of energy) and increase carbohydrate intake to roughly 50-55% of total energy intake, but with the proviso that most of the carbohydrates to be ingested at mealtime and snacks were to be from “complex” food sources. Sucrose was gradually introduced into recommendations at an intake of no more than 10% of the total energy intake. With the advent of faster acting mealtime insulins and more sophisticated insulin delivery devices (insulin pens and pumps) has come the ability to tailor precise doses to food intake preferences using carbohydrate counting and insulin dose calculators. The concept of matching insulin to carbohydrate intake, rather than food intake matched to estimated energy needs and insulin dosages, has become standard practice in T1D management. For example, self-management programs such as Dose Adjustment For Normal Eating (DAFNE) have allowed increased dietary flexibility and the potential inclusion of all foods. Thus, for several years now, low fat and high carbohydrate diets have been recommended and advocated as healthy eating for people with diabetes as well as the general population [16,17,18,19,20,21]. An analysis of the pattern of food consumption during the more recent “obesity epidemic” found that the increase in calories was almost entirely due to an increase in carbohydrate [22,23]. Today, the incidence of obesity, heart disease, and diabetes have never been higher [24], while the prevalence of obesity in T1D is now at alarming levels, perhaps in part because of the liberalisation of diet and the general sedentariness of the patient population [6].

## 3. Defining Low Carbohydrate Diets

The ADA Standards of Medical Care in Diabetes (2018) recommend 15–20% of total energy from protein, 20–35% of energy from dietary fat, which leaves a balance of 45–60% energy from carbohydrate [25]. These macronutrient guidelines are similar for children and adolescents with T1D [26]. There are no universally agreed definitions for the term “low carbohydrate diet”; however, <130 g of carbohydrate per day or <26% of total energy has been proposed [27] and these levels will vary depending on factors such as age, sex and activity level (see Table 1 and Table 2 for our working definitions). There are reports that very low carbohydrate diets (VLCD), defined as 20–50 g of carbohydrate per day [28], can lead to reductions of HbA1c as low as 5.3% in people with T1D, alongside anecdotal evidence that there are positive effects of VLCDs for glycaemic control [29,30,31]. Although there are recognised benefits of lowering HbA1c [32], there is little evidence that there is actually additional advantage of getting HbA1c lower than 7.0% in T1D with respect to overall premature mortality or morbidity risk [33], therefore these VLCDs may be unnecessarily extreme, especially if there is increased risk of hypoglycaemia or diabetic ketoacidosis (DKA) [34]. There is also evidence that people with T1D following a 1-week low carbohydrate diet (≤50 g carbohydrate per day) were at risk of impaired response to glucagon rescue during mild hypoglycaemia [35], presumably due to reduced liver glycogen content. VLCDs are generally discouraged for use by people with T1D by most certified diabetes educators and nutritionists due to somewhat theoretical concerns over potential DKA, hypoglycaemia, dyslipidaemia, nutrient deficiencies, and difficulty maintaining such diets/long-term adherence [36,37]. Regardless of whether VLCDs are effective in limiting glucose excursions and for weight control, it is likely that this approach is an unrealistic one for the majority of the T1D population, meaning compliance is likely to be low. On the other hand, it is conceivable that an LCD (i.e., 50–130 g of carbohydrate per day) may represent a realistic target that is not overly restrictive and is still likely to lead to several potential benefits, as described below. Appendix A provide examples of diets based on the different carbohydrate percentages.

It is currently unclear whether carbohydrate restrictive diets may be too restrictive for active individuals living with T1D. For the athlete with T1D (as with all athletes), nutritional requirements will be influenced by the metabolic demands of the sport that they are training for as well as the periodisation of their training and competition schedule. Indeed, it is now recognised that for the active or athletic individual, carbohydrate requirements should follow a sliding scale [38] and be adjusted day-by-day and meal-by-meal depending on the daily training and competition schedule (see Table 2 and Impey et al. [8]). Nonetheless, individuals with T1D often make claims in various social media forums and in some self-report surveys (see below) that they can still be physically active and train while adhering to a LCD. Interestingly, LaFountain and colleagues [39] found that a 12-week VLCD (≤50 g carbohydrate per day) did not compromise physical performance compared to a mixed diet in military personnel without T1D. The study also demonstrated improved insulin sensitivity and body composition following the VLCD alongside high adherence. In relation to elite endurance athletes (e.g., cyclists, runners, long distance swimmers), there is considerable confusion amongst both academic and athletic circles as to the correct terminology to use to refer to the variety of periodised carbohydrate schedules that are now commonly employed by elite athletes (see Burke et al. 2019 for a recent commentary [40]). In extreme cases of highly trained endurance athletes, carbohydrate requirements can be very high. For example, elite cyclist Chris Froome, an elite athlete who does not have T1D, was provided with a daily carbohydrate plan by his sports nutrition consultant, author JPM of this article, that provided >6500 kcal and 18 g carbohydrate per kg bodyweight during one particularly challenging solo ride over mountainous terrain during his victory in the 2018 Giro d’Italia [41]. Personal observations of the athletes, by co-author JPM of this article, demonstrate that elite professional cyclists may adjust their daily carbohydrate intake from 2–10 g per kg body mass during training depending on the goals of the specific training phase. Personal experiences with professional cyclists who have T1D (Team Novo Nordisk) by author SNS, suggest that athletes with T1D also adjust their daily intake based somewhat on daily expenditures, although these athletes must also consume carbohydrates on the ride to avoid hypoglycaemia. Future research will need to explore the effects of such high (and periodised) carbohydrate intakes on daily glycaemic profiles and long-term metabolic health in endurance athletes with T1D.

## 4. Low Carbohydrate Diets and Type 1 Diabetes

Current ADA guidelines suggest a flexible approach to carbohydrate intake matched with intensive insulin therapy [44], facilitated by programmes such as DAFNE [45]. However, there is a concern that higher caloric intakes, and potentially unhealthy eating patterns in general, are contributing to the high prevalence of obesity and the metabolic syndrome in people living with T1D [24]. Furthermore, research shows that individuals with T1D are often inaccurate when estimating the carbohydrate content of their food, particularly when the carbohydrate content is high [46,47]. This is complicated by the fact that subcutaneous insulin absorption rate may vary by as much as 30%, when insulin dosage levels are high [48,49], suggesting that lower carbohydrate meals and thus reduced insulin needs may help to reduce this variability in insulin absorption rates. A mismatch between carbohydrate absorption and insulin action typically exists after large meals in individuals with T1D, particularly with large carbohydrate meals, leading to glucose fluctuations with excessive insulin increasing the risk of hypoglycaemia [50]. As such, LCDs may enable more accurate estimates of carbohydrate content at mealtime, and as such exogenous insulin needs, to subsequently reduce glucose variability and incidence of hypoglycaemia.

A handful of studies have demonstrated promise for LCDs in people with T1D, although sample sizes have been small [29,51,52,53,54]. A recent review by Turton and colleagues [55] was unable to conclusively determine whether low vs. higher carbohydrate diets have significant effects on various T1D outcomes, such as HbA1c, total daily insulin, incidence of hypoglycaemia, and BMI, due to the large heterogeneity of the studies. Krebs et al. [51] conducted a small study to provide evidence that a LCD may be feasible for people with T1D restricted to 100 g of carbohydrate per day. Ten people with T1D were randomised to 12 weeks of either LCD or a carbohydrate counting course. Following the intervention, the carbohydrate-restricted group had a significantly reduced HbA1c from 7.9 mmol/L to 7.2 mmol/L and daily insulin use also dropped, on average, from 64 units to 44 units per day. However, continuous glucose monitor data revealed no change in glycaemic variability with the low carbohydrate intervention as compared to carbohydrate counting. Lennerz et al. [30] conducted an online survey using a social media platform to evaluate glycaemic control among adults and parents of children with T1D that reported a mean carbohydrate intake of 36±15g per day. The data collected from 316 respondents (42% were parents of children with T1D) suggested excellent glycaemic control (mean HbA1c was 5.67 ± 0.66%) with low rates of adverse events. However, there are a number of limitations of this study, including questions over how generalisable this sample population was to the rest of the population and difficulties of using self-report data [56]. For example, 36 g of carbohydrate per day is likely an underestimation of their actual carbohydrate intake from all dietary sources. Recently, Eiswirth et al. [29] conducted a case study in an adult female with T1D consuming 30–50 g of carbohydrate per day whose HbA1c reduced from 7.5% to 5.3% in four months and average daily blood glucose readings fell from 10.4 mmol/L to 6.1 mmol/L alongside reduced blood glucose variability. Importantly, Eiswirth et al. [29] found no increased episodes of hypo- or hyperglycaemia. However, this study was a case report on just one presumably highly motivated individual, following a strict insulin regime with frequent blood glucose checks, so this may not apply to the wider T1D population. Our example diets (Appendix A) demonstrate how difficult it would be for the majority of people to follow such diets safely and effectively over the long term. Ranjan and colleagues [53] investigated the short-term effects of a VLCD (≤50 g carbohydrate per day) vs. a high carbohydrate diet (≥250 g carbohydrate per day) in 10 individuals with T1D. There was no difference in mean glucose levels between the interventions but glycaemic variability was lower following the VLCD, which meant participants spent more time in euglycaemia [53]. The most promising evidence that LCDs can reduce glycaemic variability, body weight and time spent in hypoglycaemia is from a recent publication by Schmidt and colleagues [54]. Using a randomised-crossover design, the effects of 12 weeks of LCD (<100 g carbohydrate per day) were compared to 12 weeks of a high carbohydrate diet (>250 g carbohydrate per day) in 14 people with T1D [54]. There was no difference in time spent in glycaemic target range (3.9–10.0 mmol/L) between conditions; however, time spent in hypoglycaemia (<3.9 mmol/L) and glycaemic variability were lower in the LCD condition [54]. Although LCDs show some promise for people with T1D, there is a need for longer term high quality interventions. Furthermore, the effects of LCDs on metabolic response during exercise should be investigated. The following subsections will discuss the potential benefits and risks of a LCD in people with T1D. These factors are summarised in Table 3.

### 4.1. Potential Benefits of Low Carbohydrate Diets In People with Type 1 Diabetes

Obesity is increasingly common in T1D [57,58,59], with a large percentage of individuals not maintaining a healthy body mass or meeting physical activity guidelines [60,61]. Results from the Pittsburgh Epidemiology of Diabetes Complications Study showed a seven-fold increase in the prevalence of obesity from 3% to 23% over an 18-year follow up between 1988 and 2007 [58], demonstrating dramatic weight gain in this population. Large scale studies such as the Swedish National Diabetes Registry found that among the 21,000 adults with T1D on their record, obesity was significantly associated with increased risk for heart failure [62], and Price et al. [63] found that obesity was associated with retinopathy and macrovascular disease. These statistics are particularly concerning when combined with recent data showing that glycaemic control in people with T1D is getting worse [2].

In their review article, Feinman and colleagues [28] conclude that LCDs represent an effective strategy for weight loss regardless of the population, although this still needs to be tested in people with T1D. There are a number of physiological explanations to support carbohydrate restriction to manage body weight in people with T1D. First, as carbohydrate content of the diet is reduced, the relative proportion of energy derived from protein and/or fat is increased which leads to increased satiety, often resulting in reduced calorie intake [64]. Popular forms of LCDs like the Atkins diet [65] (20 g carbohydrate per day with a gradual increase to 50 g per day) or Protein Power [66] (≤100 g carbohydrate per day) put no formal limit on caloric consumption due to the assumption that greater satiety of fat and protein will control caloric intake [67].

The rise in overweight and obesity may be related to intensive insulin therapy [58] coupled with a positive energy balance. Insulin is an anabolic hormone that stimulates lipogenesis and slows basal metabolism [58,68,69], thereby resulting in increased fat accumulation. These effects are enhanced by the fact that exogenous insulin administration circulates systemically first so that it disproportionately affects muscle and adipose tissue as compared to hepatic tissue [69]. This is in contrast to healthy, non-diabetic individuals, where endogenous insulin has to first pass through the portal vein to the liver where it suppresses gluconeogenesis, and then only 40–50% of the insulin continues into the systemic circulation to act on peripheral muscle and fat tissue to increase peripheral glucose uptake and suppress lipolysis [68]. Weight gain represents a barrier to compliance of insulin therapy and diabetes control. This was highlighted by Bryden et al. [70] who followed 65 young patients with T1D from adolescence to young adulthood and found that 30% of the women admitted to under-dosing their insulin to manipulate their weight. A carbohydrate restricted diet may aid weight management in people with T1D, since caloric intake may drop and insulin dosage would likely decrease. However, there is a lack of research that has evaluated dietary approaches to optimise body weight and glycaemic control in this population [6,34,55].

Hypoglycaemia is another factor potentially impacting weight gain [6], as acute hypoglycaemia is associated with food cravings, particularly for carbohydrates to resolve the hypoglycaemia, which can lead to disinhibited eating behaviours [71]. Most notably, individuals with T1D often consume more carbohydrate than is recommended to treat hypoglycaemia and often unnecessarily treat with foods that also contain protein and fat [72]. A further danger of regularly consuming high carbohydrate snacks or meals was highlighted by the findings of Russell et al. [73], who found that healthy, normal-weight individuals that consumed an oral glucose challenge (50 g glucose) experienced a 1.5-fold greater increase in blood glucose concentration, compared to when they underwent a mixed meal challenge (21.7 g protein, 4.8 g fat, 41.0 g carbohydrate). Contrast-enhanced ultrasound showed that forearm muscle microvasulcar blood volume and flow increased following the mixed meal but decreased following the glucose challenge despite similar levels of hyperinsulinaemia between the conditions. This suggests that high glycaemic meals impair skeletal muscle microvascular blood flow, which may limit glucose disposal into the skeletal muscle. If LCDs reduce the incidence of hypoglycaemia, this would reduce the intake of corrective carbohydrates, ultimately helping to manage weight, and potentially reducing the risk of microvascular insulin resistance.

As outlined above, insulin resistance is common in people with T1D and the incidence is rising [74,75]. Insulin resistance is an independent risk factor for microvascular (neuropathy, nephropathy, retinopathy) and macrovascular (coronary and peripheral vascular disease) complications in people with T1D [74,76,77,78,79,80,81]. There are likely to be overlap in the mechanisms for insulin resistance in T1D and Type 2 diabetes, including increased intramuscular triglyceride (IMTG) content [82] and mitochondrial dysfunction [83]. However, hyperglycaemia alone is unable to explain the high prevalence of insulin resistance observed in T1D [84]. Chronic exogenous insulin use may be an important factor, as exposure to a long acting human basal insulin such as insulin detemir or glargine has been shown to result in greater insulin resistance, oxidative stress, skeletal muscle ectopic fat accumulation and mitochondrial impairments compared to hyperglycaemia alone [85]. There are studies suggesting that higher insulin dose (i.e., insulin resistance) is associated with increased mortality in people with Type 2 diabetes [86], although a cause and effect relationship remains to be established. As LCDs have been associated with reduced insulin requirements in people with T1D, likely because of reduced carbohydrate intake [51,87,88], this reduction in insulin exposure might theoretically improve insulin sensitivity based on limited animal and observational studies [89]. Further studies are required that specifically examine the effects of LCDs on insulin sensitivity via insulin clamp methods and/or meal tolerance tests in people with T1D.

### 4.2. Potential Risks of Low Carbohydrate Diets in People with Type 1 Diabetes

There are reports that VLCDs (30–40 g carbohydrate per day) in adults with T1D can lead to reductions of HbA1c to as low as ~5.5% [29,30]. However, these studies are either case reports or used very small sample sizes of presumably dedicated individuals who are not likely representative of the wider population [29] while the other report, as was mentioned above, is a large online survey of select cohorts with no controls [30]. Furthermore, and as mentioned above, there is little evidence, if any, that there is additional benefit of attaining an HbA1c lower than 7.0% on overall health or mortality [90], particularly if there is increased risk of hypoglycaemia [91], suggesting these diets may be unnecessarily extreme. VLCDs are generally discouraged in people with T1D due to concerns over potential DKA and oxidative stress, hypoglycaemia, dyslipidaemia, nutrient deficiencies, and difficulty maintaining such diets over the long term [36,37]. Of note, VLCD diets that have ketogenic effects may increase the risk of DKA in individuals taking sodium glucose co-transporter 2 (SGLT2) inhibitors since total daily insulin intake drops markedly [92] and there may be a risk of euglycaemic DKA [34]. Moreover, there may be risk for more frequent and more severe hypoglycaemia with VLCD, at least in theory. However, the increasing use of continuous glucose monitoring with alerts to warn of impending hypoglycaemia may help to facilitate a LCD or VLCD [93,94]. Elevated ketones in an insulin deficient state in people with T1D are known to elicit oxidative stress and inflammatory responses, which play a role in the development of complications [95]. Ranjan et al. raised concern for VLCDs in people with T1D in a study that demonstrated that one week of an isocaloric VLCD, defined as ≤50 g per day, reduced the treatment effect of glucagon given to treat mild hypoglycaemia [35]. Ranjan et al. suggested that the reduced glycaemic response to glucagon in the VLCD condition may be due to reduced hepatic glycogen stores. An additional concern raised by the study of Ranjan et al. [35] is that in the VLCD condition, the first glucagon bolus led to significantly higher increase in free fatty acid and ketone body concentrations compared to the high carbohydrate diet condition, suggesting that the VLCD may increase the risk of ketogenesis under certain conditions that increase glucagon levels, such as during prolonged exercise. Based on these findings, we can hypothesise that glucagon’s efficacy when used as a rescue treatment in severe hypoglycaemia would be impaired with VLCD. However, further research is needed to test this.

A second concern of LCDs or VLCDs is that as dietary carbohydrate is reduced, the intake of saturated fat will likely increase to maintain calorie intake. In a study of 11 adults with T1D on a VLCD (<55 g carbohydrate per day), 82% were found to have elevated total and LDL cholesterol and increased total:HDL ratio was evident in 64% of participants with 27% experiencing elevated serum triglycerides [31]. Similarly, 62% of participants in an observational study of a VLCD (mean intake 36 ± 15 g carbohydrate day) in adults and children with T1D [30], were shown to have dyslipidaemia (elevated total and LDL cholesterol). The impact of a LCD on blood lipids in the non-diabetic population is controversial; however, more studies provided evidence of positive [96,97,98,99,100,101,102,103,104,105] than negative effects [106]. Despite the concerns, this level of elevated cholesterol is not likely to be clinically relevant, especially if the diet leads to reduced risk of complications due to improved glycaemic control.

Another concern with a VLCD is the potential for nutrient deficiencies. Although high carbohydrate containing foods often include ingredients high in sugar or refined starches with little or no nutritional quality, carbohydrate-rich foods also include a wide range of nutritionally valuable options such as fruits, vegetables, whole grains, milk, yogurt, and legumes. Omitting these later foods may not be ideal from a healthy eating perspective, since they often contain valuable nutrients and fibre. Dairy consumption, for example, has been shown to have several health promoting effects in diabetes, perhaps via probiotic actions [107]. However, although yogurt consumption, as one example, can be beneficial to health, products can vary widely in sugar content and it is common for the carbohydrate content of low fat yogurts to be over 10% due to addition of low-cost fructose syrup [108]. It is also worth emphasising the dangers of regular consumption of high fructose syrup in some yogurts and fizzy drinks, particularly in children, for the risk of obesity, insulin resistance, cardiovascular disease, fatty liver disease and dental carries [109,110,111]. Moreover, a LCD does not always distinguish between higher quality and lower quality food choices. For example, analysis revealed that the *Atkins for Life* VLCD (20–40 g carbohydrate per day) was insufficient in delivering Reference Daily Intake for 12 of the 27 micronutrients [112]. In a recent hypothetical case study design, a low carbohydrate (<130 g per day), high fat diet was found to meet all minimum micronutrient thresholds, apart from iron in the female meal plans, which achieved 86–98% of the threshold [113]. There is also evidence to suggest that a high fat diet, as a function of a VLCD, can cause iron deficiency: Leow et al. [31] reported that 3 out of 7 men on a VLCD had low haemoglobin. Therefore, evaluation and monitoring of vitamin and mineral status in individuals with T1D following a LCD should be encouraged and multi-vitamin supplements may be recommended to reduce the risk of deficiencies. Of particular concern is the effect of a VLCD on bone health with often inadequate calcium and vitamin D intakes [112]. This is particularly relevant in T1D, since suboptimal glycaemic control appears to be associated with a reduction in bone mineral density [114,115] and nearly 20% of people with T1D between the ages of 20–56 years meet the criteria for being osteoporotic [116]. Although there is limited data on the long-term effects of a LCD in people with T1D, following a ketogenic diet has been shown to result in reduced bone mineral content in children with epilepsy [117]. Furthermore, despite significant evidence of the beneficial role of low glycaemic index carbohydrates in the management of postprandial glycaemia in T1D [118], the current trend continues to emphasise quantity of carbohydrate in the diet over quality. This may be an oversimplification as the restriction of poor quality, refined carbohydrates high in sugar and low in dietary fibre is beneficial for overall health, but some high-quality carbohydrates are likely to be health promoting, since they often contain essential micronutrients and other potentially beneficial nutraceuticals. Dietary guidelines including the ADA’s Standards of Medical Care in Diabetes encourage nutrient-dense carbohydrate sources that are high in fibre, including vegetables, fruits, legumes, whole grains, as well as dairy products [119].

A nutrient-rich diet that meets the energy, vitamin and mineral requirements is essential for normal growth and development, and failure to meet these needs may be particularly detrimental in children [26]. de Bock and colleagues published a series of case studies suggesting that carbohydrate restriction in growing children can lead to anthropometrical and maturational deficits and increase cardiovascular disease risk profile due to increased fat intake [36]. However, if compensatory energy is not provided in the form of increased fat and protein intake, carbohydrate restriction will result in total energy reduction with weight loss and potential negative impact on growth in children and adolescents. The role of a specialist dietitian in assessing the overall diet to ensure all requirements are being met is thought to be essential for people living with T1D [34]. Readers are referred to the latest ISPAD Clinical Practice Consensus Guidelines on nutritional management in children and adolescents with diabetes for additional information on the importance of appropriate energy intake to achieve optimal body weight, growth and development [26]. In general, VLCDs are not advised for youth with T1D and it should be highlighted that the timing and amount of carbohydrate intake around exercise and physical activity presents additional challenges to the child and their family. Current strategies recommend additional carbohydrate-based snacking for prolonged activities to help minimise hypoglyacemia risk [120].

The management of T1D places tremendous emphasis on food selection and portion size. When an additional dietary restriction is added, the risk of food pre-occupation often increases. Compared to people who do not live with diabetes, individuals with T1D have a greater risk of developing eating disorders or other psychological disorders [121,122]. In addition, long-term adherence to LCDs has shown varying results [30,51,123]. Very few long-term studies in T1D and LCDs exist; however, Nielson et al. [87] reported results from a retrospective audit of dietary adherence (<75 g carbohydrate per day) and found a dropout rate of 52%. Sustainability is a critical component to any dietary recommendation. Dietary restrictions may be difficult to manage in the short and long term given our cultural norms, the role of food in celebrations, and other situations such as restaurant meals and traveling. An LCD diet may be socially isolating and increase the risk of developing disordered eating behaviours. This social isolation may help to explain why social media pages touting VLCD are prevalent even in the T1D community.

## 5. Staying Active on a Low Carbohydrate Diet

Exercise presents particular metabolic challenges for people with T1D as it significantly increases the risk of hyper- and hypoglycaemia, depending on multiple variables [124]. Maintaining blood glucose concentrations within an acceptable range before, during, and after a bout of exercise, to achieve optimum performance and training adaptation is a balancing act between the correct insulin dosing strategy and carbohydrate intake. Control of carbohydrate metabolism during endurance exercise depends on the availability of endogenous and exogenous carbohydrate stores, the abundance and activity of transport proteins involved in transporting substrates across plasma and mitochondrial membranes, and the activity of enzymes involved in metabolic pathways. Exercise intensity, duration, nutritional status, and training status all have an impact on substrate utilisation during exercise and the risk for glycaemic disturbance for people with T1D. For the active patient with T1D, additional factors such as on-board insulin concentration and the factors shown in Table 4 are also important in predicting the impact of exercise on glycaemia. Strategies to optimise the timing and amount of carbohydrate intake may have a large impact on performance, blood glucose management, and weight control.

## 6. Carbohydrate Restriction Strategies for the Endurance Athlete with Type 1 Diabetes—Practical Considerations

Despite low carbohydrate training being one of the most widely debated topics amongst athletes, coaches and sport scientists, there is very little published research specific to athletes with T1D. Carbohydrate requirements depend on the training status and the event that the individual is training for. Additional factors that can influence blood glucose concentrations in athletes with T1D make the estimate of carbohydrate requirements complex (Table 4). A plethora of scientific evidence in people without T1D suggests that a high carbohydrate diet supports endurance performance in athletes and non-athletes [125,126,127,128,129,130]. However, periodically undertaking endurance training with reduced carbohydrate availability, but then performing competition without carbohydrate restriction (i.e., ‘training low’ but ‘competing high’) has been shown to promote superior training adaptations in skeletal muscle when compared with high carbohydrate availability in athletes without diabetes (reviewed in [8]). The majority of studies suggest increased cell signalling, gene expression and training induced increases in the oxidative capacity of skeletal muscle (Figure 1). However, these adaptations do not always translate to improved exercise performance [12,131,132,133,134].

Developing a nutritional strategy for a high level or elite athlete with T1D is extremely complex due to the number of factors that need to be considered alongside the already difficult task of managing blood glucose levels in everyday life (Figure 2). Athletes need adequate carbohydrate to fuel each training session so that they are completed at an intensity necessary to elicit adaptation, and there may often be the requirement to reduce fat mass and retain or increase lean mass in preparation for competition [43,143]. The athlete and their coach must also be aware of training load and competition schedule, which means carbohydrate intake is likely to vary from day to day. Adequate carbohydrate intake is needed for restoration of muscle glycogen stores between sessions, which is especially important for athletes undertaking long duration or high intensity training sessions and competing more than once within a short space of time [144,145]. Weight control can be particularly challenging for athletes with T1D as high levels of insulin will increase carbohydrate needs during exercise and suppress the use of endogenous fuel stores. This is crucial to take into account when competing in certain events such as long distance running; cycling where the aim is to improve power to weight ratio; or sports where certain weight categories need to be met such as boxing, martial arts, and horse racing.

A chronically low carbohydrate diet, as discussed in the previous sections, is likely not suitable in high level athletes with T1D due to the energy required to fuel their heavy training load. However, there are a number of ‘train low, compete high’ strategies that have been investigated previously in people without T1D [8] including twice per day training, fasted training, post exercise carbohydrate restriction, and sleep low, train low strategies; some of which may or may not be beneficial for people with T1D (Figure 3). The athlete with T1D needs to ensure that training intensity is not compromised while creating a metabolic milieu conducive to facilitating endurance phenotype alongside maintaining a blood glucose concentration that is safe and does not impair performance. As highlighted by Bartlett et al. [7], training with low glycogen stores has limitations in athletes without T1D. For example, performing exercise in a low carbohydrate state may be difficult to maintain training intensity, thus impairing adaptation. Furthermore, exercising regularly without or with low exogenous carbohydrate may impair the ability to subsequently oxidise exogenous carbohydrate when intake is increased [154], which may impact performance during competition. 

Finally, any augmented muscular adaptations using these ‘train low’ strategies may have additional metabolic benefits for people with T1D due to improved overall skeletal muscle health. This is important, as skeletal muscle health has been identified to be impaired in people with T1D (e.g., mass, function, metabolism) and therefore represents a therapeutic target [83,155,156,157]. Monaco and colleagues [157] outlined their hypothesis that T1D is a form of accelerated ageing in skeletal muscle due to impairments in mitochondrial structure and function, potentially accelerating the decline in muscle health. Therefore, improvements in skeletal muscle health due to augmented mitochondrial protein expression may be particularly important for athletes with T1D, irrespective of any performance benefits for improved glycaemic control and metabolic health, which would suggest that investigating the effects of train low strategies is worthwhile. For example, augmented cell signalling leading to improved mitochondria structure and function may be beneficial for performance and/or metabolic health seeing as mitochondria structure is impaired in people with T1D [83]. However, this is speculative so needs to be tested properly. 

Adaptations to exercise under carbohydrate restricted conditions are yet to be tested in people with T1D, therefore the majority of this section is based on extrapolation of mechanisms that operate in people without T1D. The following sections will focus on endurance athletes, although the authors acknowledge that this is still a simplistic approach as even among elite athletes, training loads can vary as much as 10–12 h per week with individual training sessions lasting between 1–6 h for cyclists and triathletes. If research in this area is to be conducted, it is essential that glycaemic control during and after exercise and overall metabolic health are given just as much attention as performance outcomes when studying dietary interventions in individuals with T1D. It is important to conduct studies characterising the 24-h glucose profiles, changes in lipid metabolism, body composition and HbA1c following periodisation training strategies in people with T1D.

### 6.1. Twice Per Day Training

Twice per day training is another training approach that is designed such that the athlete completes a morning training session followed by several hours of reduced carbohydrate intake so that the second training session of the day is commenced with reduced muscle glycogen (i.e., training glycogen depleted). In individuals without T1D, 3–10 weeks of twice per day training has been shown to increase oxidative enzyme activity [131,132,158], whole body lipid oxidation [131,132] and intramuscular lipid utilisation [131], aerobic capacity [158] and performance [159] vs. once per day training. The first study to investigate this approach was by Hansen et al. [158] who used a one-legged kicking model to compare once per day training against training twice per day, every other day. This meant that the second exercise bout in the twice per day condition was performed with low muscle glycogen. These positive results of twice per day training were subsequently shown on a whole-body level by others [131,132].

Although, yet to be tested in a research environment, twice per day training is likely to be technically challenging for people with T1D. Firstly, antecedent exercise blunts the counter-regulatory responses during subsequent exercise which would increase the risk of hypoglycaemia [160]. Secondly, twice per day training would require regular glucose monitoring throughout training days and increased vigilance overnight to prevent nocturnal hypoglycaemia. Currently there is no evidence on whether multiple exercise sessions per day increase the risk of nocturnal hypoglycaemia compared to a single session. However, under the right conditions, athletes may be able to train twice daily if they consume some carbohydrates during the second training session to avoid hypoglycaemia and use continuous glucose monitoring. 

### 6.2. Fasted Exercise in Athletes with Type 1 Diabetes

Fasted exercise, whereby breakfast is consumed after a morning training session, is a simpler model of train low that may lead to superior metabolic adaptations to training in the fed state (see Wallis and Gonzalez [161] for a recent review of non-diabetes studies). Exercising in the overnight fasted vs. fed state in people without diabetes has been linked to a number of responses that may translate to long-term improvements in lipid and glucose metabolism [162]. During fasted exercise, there is increased fat utilisation, improved plasma lipid profiles and enhanced activation of the molecular signalling pathways, leading to increased mitochondrial content and capacity to oxidise fat while supressing glucose metabolism compared to fed exercise [163,164,165]. The kinases involved are 5′ adenosine monophosphate-activated protein kinase (AMPK) activity, CaMK and p38MAPK [166]. During exercise, these kinases translocate from the cystosol in to the nucleus where they activate transcription factors controlling the expression of nuclear genes known to control mitochondrial biogenesis and genes encoding for fatty acid transport into skeletal muscle, the capacity of the carnitine shuttle and the β-oxidation enzymes [164]. Feeding before exercise lowers post-exercise adipose tissue gene expression of pyruvate dehydrogenase kinase isozyme 4, adipose triglyceride lipase, hormone-sensitive lipase, GLUT4 and insulin receptor substrate 2 compared to the fasted state [163]. Stocks et al. [162] found that undertaking moderate- to high-intensity steady-state exercise (70% W_max_) in the fasted state increased fatty acid availability, augmented AMPK^Thr172^ phosphorylation and increased PDK4 mRNA expression compared to the same exercise bout in the fed state. Carbohydrate intake before and during exercise stimulates a contribution of blood glucose to the metabolic substrate pool fuelling muscle and inhibits fat oxidation due to higher glycolytic flux and pyruvate oxidation by the pyruvate dehydrogenase complex [167,168,169,170,171,172]. This leads to attenuated lipid utilisation during exercise [10,168], mainly via higher insulin concentrations, which inhibits lipolysis both of the adipose tissues stores (reducing plasma NEFA concentrations) and the IMTG stores [173,174]. The high insulin concentration, as typically occurs in the post meal state particularly if carbohydrates are consumed, will also suppress liver glycogen breakdown and gluconeogenesis, meaning that glucose production by the liver will not match glucose oxidation by the muscle, so blood glucose concentration will fall [175]. In contrast to fed state exercise, during fasted exercise there is increased mobilisation of triglyceride reserves from adipose tissue and decreased re-esterification of NEFAs and insulin concentrations will be low. This leads to an increase in concentrations of circulating NEFA in the plasma and consequently more lipid provision to the muscles for oxidation [176,177]. The low insulin levels will also lead to reduced blood glucose oxidation and muscle glycogen breakdown, as well as greater liver glucose production through glycolysis and gluconeogenesis. Importantly, individuals with T1D can slightly modify the fuel selection during exercise based on when the exercise is performed relative to their bolus insulin administration and how much basal and bolus insulin is in circulation at the time of exercise. For example, reducing or withholding insulin can increase lipid mobilisation, and reduce the body’s reliance on carbohydrate as fuel [178]. However, withholding insulin can also dramatically increase glycaemia and ketone production during exercise [179]. 

There is emerging evidence in people without T1D that regularly training in the fasted state is beneficial for metabolic adaptations [10,11,12]. Van Proeyen et al. [161] investigated the effects of six weeks of supervised endurance training under tight dietary control on metabolic adaptations to exercise where half of the participants consistently trained in the fasted state and the other half were fed carbohydrate before and during each training session. Muscle histology showed that fasted training enhanced the contribution of intramyocellular lipids (IMCL) to energy provision, but this did not occur in the fed group. The findings suggest that fasted training induced adaptations to muscle cells to shift fat fuel selection towards a higher fraction of IMCL utilisation. However, although this study suggested that training in the fasted state is a more potent stimulus than fed training to enhance muscular oxidative capacity, the enhanced oxidative capacity in fasted training did not translate to better performance in the 60-min time trial. Specifically, for people with T1D, these findings suggest the importance of reducing insulin dose prior to exercise to increase the contribution of lipid oxidation to fuel the exercise bout which may help to reduce fat mass and attenuate a reduction in blood glucose concentration. To reduce insulin effectively in those individuals on insulin pumps, aggressive basal rate reduction (by 80%) ~90 min pre-exercise is required and the effects in lipid utilisation may only be marginal [180].

Interestingly, in the study by van Proeyen and colleagues [12], fasted training but not fed training prevented the exercise-induced drop in blood glucose concentration during a bout of fasted exercise that was performed post training. The authors [12] speculated that regular exercise in the fasted state may stimulate adaptations in the liver to facilitate glucose production via gluconeogenesis. If this adaptation also occurs in people with T1D, regular fasted training may be a means to attenuate the decline in blood glucose during moderate-intensity exercise and therefore reduce the risk of hypoglycaemia. On the other hand, fasted training may promote a consistent rise in blood glucose level if the exercise intensity is too vigorous [181,182]. Another interesting finding was by Stannard et al. [11], who showed that training five times per week for four weeks in an overnight fasted state enhanced muscle glycogen storage more than fed training. Resting muscle glycogen stores increased just 3% in the fed training condition but 55% in fasted state, suggesting enhanced training adaptation. If these adaptations also occur in individuals with T1D, regular fasted training may improve metabolic adaptation and also improve glycaemic control and reduce the risk of hypoglycaemia over the 24-h period. The effects of regular long-term training in the fed vs. fasted state has not been tested in people with T1D. 

In those living with T1D, blood glucose concentration has been found to be more stable following fasted exercise performed in the morning [183], in contrast with the declines in blood glucose found during later day (fed state) aerobic exercise [184,185,186,187,188,189,190,191] and high-intensity interval training [184,185,187,190,191]. This may be due to a greater reliance on fat oxidation during fasted exercise that may offset the fall in glycaemia and reduce the risk of hypoglycaemia. More brief sessions of intensive high-intensity interval type exercise may even promote a significant rise in glycaemia that requires insulin correction before the first meal [192]. However, it is unclear if exercising in the fasted state would be beneficial when undertaking training sessions longer than 30 min, which makes this information impossible to relate to elite athletes that commonly train for over 6 h. There are several possible explanations for this phenomenon of fasted exercise. Firstly, lower circulating insulin during fasted exercise attenuates the suppression of hepatic glycogenolysis, and consequently increases blood glucose during exercise. Secondly, individuals with T1D may experience the “dawn phenomenon” [193] when the exercise is performed in the morning, whereby there is an early morning rise in blood glucose possibly due to greater circulating growth hormone [194,195,196]. While these theories remain unconfirmed, it can still be suggested that those struggling with hypoglycaemia during exercise, and/or those trying to avoid additional carbohydrates to aid weight management, may have greater success with early morning/fasted exercise than they would with exercise later in the day. In general, these studies suggest the potential for overnight fasted exercise to promote greater benefits to metabolic health than regular fed exercise; however, research is still limited in this area [161], especially in those with T1D. The authors suggest that fasted exercise represents a promising strategy for people with T1D, as it may reduce the risk of exercise induced hypoglycaemia alongside improved adaptation, and should be explored further.

### 6.3. Sleep Low, Train Low

‘Sleep low, train low’ involves performing a bout of exercise in the evening, restricting carbohydrate intake overnight, and then exercising in the fasted state the following morning [197]. The ‘low’ in this context refers to muscle glycogen stores in this context and not hypoglycaemia per se. The proposed benefit of this model is that the accumulated total time spent in a state of reduced muscle glycogen can be as long as 12–14 h, depending on the timing and duration of the training sessions and sleep period [8]. Studies using the sleep low, train low model have been shown to enhance activation of AMPK, p38 MAPK and p53 signalling [198]. Marquet and others [199,200] investigated the effects of 1–3 weeks of sleep low training in elite level cyclists and triathletes without diabetes on a simulated triathlon race performance. They found increased cycling efficiency, 20 km time trial performance, and 10 km running performance compared to traditional high carbohydrate training. 

There are no trials that have investigated sleep low, train low in people with T1D. However, regardless of whether sleep low, train low augments training adaptation, this strategy is unlikely to be a viable option for people with T1D due to the high risk of nocturnal hypoglycaemia.

## 7. Conclusions and Future Directions

The current trend of increasing obesity rates and impaired metabolic control in people with T1D, despite improvements in insulins and technology [2], suggests novel, achievable lifestyle changes are needed. Sensible LCDs consisting of <130 g carbohydrates per day may represent a strategy to improve glycaemic control and metabolic health in people with T1D. The daily demands placed on an individual with T1D to manage their blood glucose are complex, therefore, strategies must be achievable, increase quality of life and ensure enjoyment of food while taking social and psychological factors into account in order to be successful. Nutritional guidance for people with T1D must be based on appropriate scientific evidence using sound methodologies. The limited research that does exist suggests that LCDs represent an effective strategy to improve glycaemic control and metabolic health in people with T1D and that this would increase the time in target glycaemia due to the reduced risk of hypo and hyperglycaemia. There is also strong reason to believe that LCDs would improve body composition and lower insulin requirements, which is beneficial for overall cardio-metabolic health. However, there is a clear lack of well-conducted randomised controlled trials to conclusively state the degree of carbohydrate restriction that is safe, feasible, and effective over the long term in people with T1D. Studies investigating LCDs have been mostly cross-sectional with a lack of control groups and the case reports in this area are from highly motivated self-selected individuals who follow intensive insulin management strategies and are not representative of the general population with T1D. Many LCD studies have relied on self-report dietary information, which has its own inherent challenges. For the recreationally active and elite level athlete with T1D, the research in this area is even sparser. Research conducted on people without T1D suggests that certain ‘train low’ strategies are beneficial for metabolic adaptation compared to training with high carbohydrate availability. However, these practices are yet to be tested in the athlete with T1D and some strategies represent additional challenges that make this unsuitable for the athlete with T1D, regardless of any potential effectiveness for performance (see Table 5 for further research questions). Clearly the use of LCDs is an intriguing area of research that warrants greater attention in people with T1D.

## Figures and Tables

**Figure 1 nutrients-11-01022-f001:**
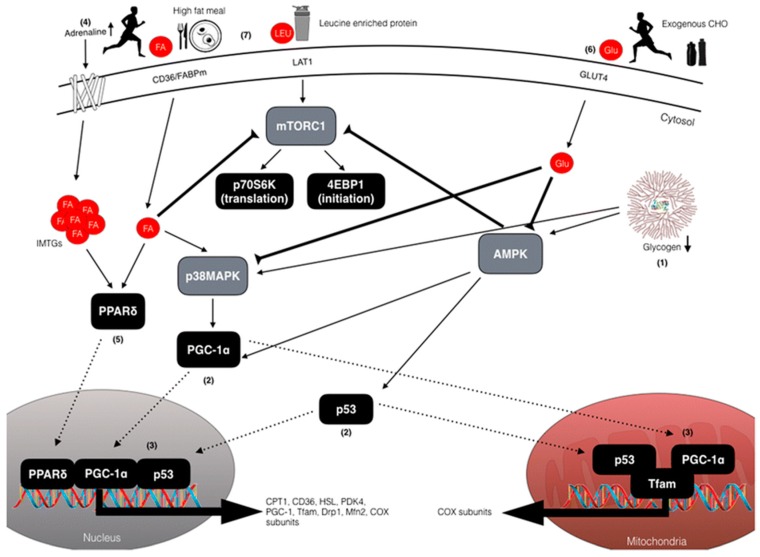
Schematic overview of the exercise-nutrient-sensitive cell signalling pathways regulating enhanced mitochondrial adaptations associated with training with low carbohydrate availability in people without Type 1 diabetes (T1D). Adapted with permission from [8]. (1) Reduced muscle glycogen enhances 5′ AMP-activated protein kinase (AMPK) and p38 mitogen-activated protein kinase (p38MAPK) phosphorylation resulting in (2) activation and translocation of peroxisome proliferator-activated receptor gamma coactivator 1-α (PGC-1α) and p53 to the mitochondria and nucleus. (3) Upon entry into the nucleus, PGC-1α co-activates additional transcription factors (i.e., NRF1/2) to increase the expression of cytochrome c oxidase (COX) subunits and mitochondrial transcription factor A (Tfam), as well as autoregulating its own expression. In the mitochondria, PGC1α co-activates Tfam to coordinate regulation of mtDNA, and induces expression of key mitochondrial proteins of the electron transport chain; e.g., COX subunits. Similar to PGC-1α, p53 also translocates to the mitochondria to modulate Tfam activity and mtDNA expression, and to the nucleus where it functions to increase expression of proteins involved in mitochondrial fission and fusion (i.e., dynamin-related protein 1 and mitofusion-2) and electron transport chain proteins. (4) Exercising in conditions of reduced carbohydrate availability increases adipose tissue and intramuscular lipolysis via increased circulating adrenaline concentrations. (5) The resulting elevation in free fatty acids (FFA) activates the nuclear transcription factor, peroxisome proliferator-activated receptor δ, to increase expression of proteins involved in lipid metabolism. (6) However, consuming pre-exercise meals rich in carbohydrates and/or carbohydrate during exercise can downregulate lipolysis (thereby negating FFA-mediated signalling), as well as reduce both AMPK and p38MAPK activity, thus having negative implications for downstream regulators. It is important to note that these signalling responses are likely different in athletes with T1D due to dependence on exogenous insulin that will suppress lipolysis and activation of the PPARs, which in turn may impair adaptations in mitochondrial proteins and oxidative enzymes following exercise. This may suggest a mechanism for enhanced adaptation with fasted exercise in athletes with T1D, but this hypothesis needs to be tested. (7) High fat feeding can also modulate PPARδ signalling and upregulate genes with regulatory roles in lipid metabolism and downregulate carbohydrate metabolism.

**Figure 2 nutrients-11-01022-f002:**
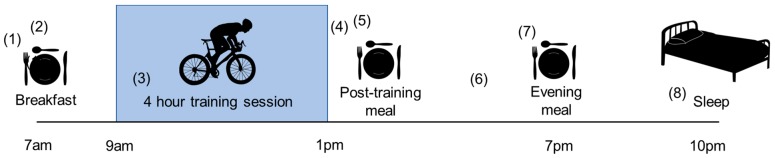
An example day in the life of a high-level endurance athlete living with Type 1 diabetes. This is a simplified schematic to demonstrate the complexity of managing the many factors that the athlete must take into account to achieve optimal nutrition for training, recovery and to spend as long as possible in the target glycaemic range. These factors will vary between and within athletes, depending on training schedule and nutrition requirements. (1) Upon waking, liver glycogen levels will be low, therefore breakfast should be planned with the morning training session in mind to ensure the athlete is fuelled and blood glucose levels are in the target range to complete the session. (2) Reduced insulin dose with breakfast, by 25–75%, is an important consideration in order to minimise a drop in blood glucose concentration during training. (3) In ride nutrition needs to be planned in accordance with their workload, pre-exercise blood glucose concentration, and training conditions (e.g., if session is at altitude and/or high ambient temperature). The aim is to ensure that the athlete is not at risk of hypoglycaemia or hyperglycaemia while taking on carbohydrates to maintain carbohydrate availability to promote training intensity (carbohydrate intake can be low, simply to reduce risk for hypoglycaemia [146], moderate if insulin dose reductions are performed [147] or can be as high as 60–75 g per hour to maximise performance [148]). Hydration is also important, with fluid consumption matched to water loss from sweat and respiration. (4) Post training nutrition is important to capitalise on glycogen storage, which is particularly important if they plan on training on the same or following day. (5) The athlete may consider adjusting their insulin dose post-training depending on their blood glucose levels. Typically less basal insulin is required for active days [149]. (6) Snacks containing carbohydrates may be consumed throughout the rest of the day to maintain fuelling and to prevent hypoglycaemia. (7) The macronutrient content of the evening meal and insulin bolus is important to ensure refuelling glycogen stores following the training session. (8) It is also important that the athlete gets enough undisturbed sleep to recover, which can be a challenge if glucose control is compromised overall [150] and because late day activity may compromise sleep quality since nocturnal hypoglycaemia risk increases [151]. The risk of nocturnal hypoglycaemia can be increased following certain training sessions [151,152,153]. Therefore, the athlete should consider adjusting their insulin dose overnight to increase the time in target and reduce the risk of disrupted sleep. Continuous glucose monitoring is a useful tool for the athlete to monitor difficult areas throughout their daily schedule to reduce the time spent in hyper or hypoglycaemia, and to help ensure that their nutrition is adequate for their training schedule.

**Figure 3 nutrients-11-01022-f003:**
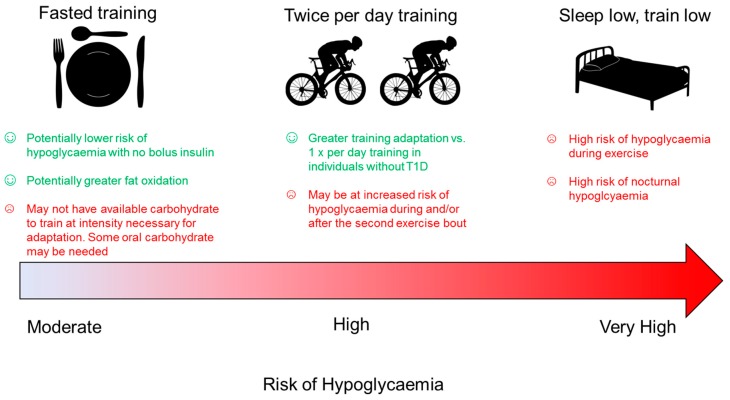
Summary of presumed hypoglycaemia risk and benefit to training adaptation in people with Type 1 diabetes using ‘train low’ strategies.

**Table 1 nutrients-11-01022-t001:** Suggested definitions for different carbohydrate diets based on values from Feinman et al. [28] and Seckold et al. [34].

Diet	Recommendation
Very low carbohydrate diet	20–50 g per day or <10% caloric intake or <1 g/kg bodyweight/day
Low carbohydrate diet	<130 g per day or <26% total energy intake or <3 g/kg bodyweight/day
Moderate carbohydrate diet	26–45% of total energy intake or 3–6 g/kg bodyweight/day
High carbohydrate diet	>45% of total energy intake or 7–8 g/kg bodyweight/day
ADA guidelines [42]	45–60% total energy intake from carbohydrate

**Table 2 nutrients-11-01022-t002:** Guidelines for carbohydrate intake by endurance trained athletes, adapted from Burke et al. [43] and the position of the Academy of Nutrition and Dietetics, Dietitians of Canada, and the American College of Sports Medicine guidelines [38].

Level of Activity	Carbohydrate Targets
Light (low intensity or skill-based activities)	3–5 g/kg bodyweight/day
Moderate (approximately 1 h per day)	5—7 g/kg bodyweight/day
High (e.g., 1–3 h moderate to high-intensity exercise)	6–10 g/kg bodyweight/day
Very high (e.g., >4/5 h of moderate to high-intensity exercise)	8–12 g/kg bodyweight/day
Extreme (e.g., elite cycle competition)	>12 g/kg bodyweight/day

N.B. Timing of intake of carbohydrate over the day may be manipulated to promote high carbohydrate availability for a specific session by consuming carbohydrates before or during the session, or during recovery from a previous session.

**Table 3 nutrients-11-01022-t003:** Potential benefits and negatives of people with Type 1 diabetes following a low carbohydrate diet.

Potential Pros of Low Carbohydrate Diets	Potential Cons of Low Carbohydrate Diets
Reduce HbA1c	Risk of nutrient deficiencies
Reduced glycaemic variation Strategy for weight loss	Potential risk of diabetic ketoacidosis
Decreased total daily insulin dose	Reduced treatment effect of glucagon during hypoglycaemia
	Increased saturated fat intake to maintain calorie intake
	Risk of pre-occupation with food and eating disorders
	Difficulty with sustaining low carbohydrate diets
	Possible maturational deficits in children

**Table 4 nutrients-11-01022-t004:** Factors to consider when calculating carbohydrate intake requirements for the active patient with Type 1 diabetes (T1D).

Factor	Comments	Implications for the athlete with T1D
Exercise modality and protocol	Exercise modality, duration and intensity can all affect muscle glucose uptake and both liver and muscle glycogenolysis.	Carbohydrate requirements will be greater with greater training loads. The type of exercise influences the change in glycaemia [124,135].
Environmental conditions	Training/competing at high temperatures and/or at altitude increases the risk of hypoglycaemia [136].	Extra consideration is needed, especially if they are accustomed to lower temperatures.
Antecedent hypoglycaemia and/or moderate intensity exercise	Counterregulatory responses may be impaired during subsequent exercise bouts and increase the risk of hypoglycaemia [137,138].	Following recent hypoglycaemia, carbohydrate requirements during subsequent training sessions may be greater than usual.
Pre-exercise blood glucose levels	There is evidence that blood glucose drops more when starting exercise with higher blood glucose concentration [139].	If blood glucose is elevated, carbohydrate feeding may need to be delayed until blood glucose has lowered. However, when pre-exercise blood glucose is low, high glycaemic index carbohydrate may need to be consumed.
Time of day	Exercising late in the afternoon may increase the risk of nocturnal hypoglycaemia [140]. Early morning exercise may reduce risk of hypoglycaemia due to the ‘dawn effect’.	The athlete may require more vigilance after an afternoon exercise session to reduce the risk of nocturnal hypoglycaemia.
Hormonal factors	Menstrual cycle phase in women [141] and possibly competition stress [142] (i.e., insulin resistance during early luteal phase and a rise in glucose level during competition stress associated with cortisol and/or catecholamines)	Adrenaline release before competition may cause blood glucose levels to rapidly rise. Blood glucose responses during training may be very different during high stress competition settings.

**Table 5 nutrients-11-01022-t005:** Further areas for research specific to athletes and carbohydrate intake in athletes with T1D.

How do athletes with T1D periodise their training and diet over a training season, and is this the optimal strategy?
What are the effects of long-term fasted exercise training in athletes with T1D? Are there benefits and/or disadvantages?
Does a high frequency of heavy training sessions lead to an accumulative increase in the risk of hypoglycaemia?
Are T1D athletes refuelling adequately during training and competition for subsequent exercise?
What are the barriers to exercise for high level athletes with T1D?
What is the best strategy to periodise training and diet over a training season in athletes with T1D?
Are athletes with T1D in a chronic state of low energy availability?
What are the acute and chronic effects of hyperglycaemia during exercise in athletes with T1D?
Will future use of artificial pancreas technology during endurance exercise lead to better glucose homeostasis in athletes with T1D?

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
