# Peer review of "Carbohydrate Restriction in Type 1 Diabetes: A Realistic Therapy for Improved Glycaemic Control and Athletic Performance?"

_nutrients, 2019, doi:10.3390/nu11051022_

Round 1
Reviewer 1 Report
The authors are to be commended for this comprehensive and detailed review of an important albeit under-studied field of Research. I have one essential suggestion: while a lot is said about the quantitative restriction of carbohydrates in this context, I would encourage the authors to add a para on qualitative aspects (ie different types of carbs and their impact on the proposed mechanisms, and potentially also the role of different types of Proteins/amino acids, and Lipids in These types of diets). Apart from These aspects I deem this review highly comprehensive and informative.
Author Response
Response to Reviewer 1 Comments
We thank the Reviewer for their helpful comments and the Editor for providing us with the opportunity to revise the paper. We have adapted the manuscript in line with the Reviewers’ comments. The responses to individual comments are described below in red and revisions made in the manuscript are also in red.
Point 1: The authors are to be commended for this comprehensive and detailed review of an important albeit under-studied field of Research. I have one essential suggestion: while a lot is said about the quantitative restriction of carbohydrates in this context, I would encourage the authors to add a para on qualitative aspects (ie different types of carbs and their impact on the proposed mechanisms, and potentially also the role of different types of Proteins/amino acids, and Lipids in These types of diets). Apart from These aspects I deem this review highly comprehensive and informative.
Response 1: Thank you for your comments. In regard to your suggestion we have expanded paragraph 3 of section 4.2 where we had some discussion on nutrient deficiency and dietary quality. We added the following sentence to emphasise dangers of high fructose syrup in some foods on Lines 373-375 (highlighted in red):
‘It is important to emphasise the dangers of regular consumption of high fructose syrup in some yogurts and fizzy drinks, particularly in children, for the risk of obesity and dental carries [103].’
We have also added the following section regarding glycaemic index of carbohydrates on lines on Lines 391-400 (highlighted in red):
‘Furthermore, despite significant evidence of the beneficial role of low glycaemic index carbohydrates in the management of postprandial glycaemia in T1D [118], the current trend continues to emphasise quantity of carbohydrate in the diet over quality. This may be an oversimplification as the restriction of poor quality, refined carbohydrates high in sugar and low in dietary fibre is beneficial for overall health but some high-quality carbohydrates are likely health promoting since they often contain essential micronutrients and other potentially beneficial nutraceuticals. Dietary guidelines including the American Diabetes Association's Standards of Medical Care in Diabetes encourage nutrient-dense carbohydrate sources that are high in fibre, including vegetables, fruits, legumes, whole grains, as well as dairy products [119].’
We feel that to include any additional information on the quality of fatty acids and protein consumed goes beyond the scope of our review so have not added this.
Reviewer 2 Report
The manuscript deals with a very interesting research topic. It analyses the impact of low carbohydrate diet in type 1 diabetes on improving glycaemic control, with special attention to the potential for restricted carbohydrate diets in athletes with T1D. The manuscript is structured in a coherent manner, however, I do have several comments and recommendations:
- Pages 1-2. The introduction is too long and multithreaded; this section could be shortened.
- The Authors of this manuscript should present and discuss the ISPAD Clinical Practice Consensus Guidelines 2018 in the context of opportunities and limitations of low carbohydrate diet for youth with type 1 diabetes. Please include the references: Nutritional management in children and adolescents with diabetes. Pediatr Diabetes. 2018, 19 Suppl 27:136-154; and Exercise in children and adolescents with diabetes. Pediatric Diabetes. 2018, 19 Suppl 27:205-226. The authors refer to youth with type 1 diabetes in the introduction to the manuscript.
- The individual sections in the manuscript, except for the introduction should be numbered (sections and subsections). Please add such numbering.
- Page 2 (line 87): Please present the aim of this review in numbered points, in the order corresponding to the body of the manuscript.
- Page 2 (line 94): ‘History of carbohydrate restriction in people with type 1 diabetes’ – this section could also be shortened.
- Page 5 (line 189-223): ‘Low carbohydrate diets and type 2 diabetes’ – I suggest deleting the subsection since the aim of the manuscript as identified by the Authors refers to type 1 diabetes only.
- Pages 7-9 (line 265-280): Please move Table 3 to Supplementary Table S1 (Supplementary material). Page 6 (line 249): Please refer to Supplementary Table S1 instead.
- Page 10 (line 347): ‘Potential risks of low carbohydrate diets in people with type 1 diabetes’ – reference to Table 4 is missing from this section.
- Page 14 – Please move Table 5 to page 12, above Figure 1.
- Page 16 (line 523) – Error in reference number; In References (page 27) – incomplete information for reference no. 79.
- Page 23 - Table 6 - Further areas for research specific to athletes and carbohydrate intake in athletes with T1D – Please add in separate subsections: Conclusions and Future Directions.
- Please attach a list of abbreviations.
Author Response
Response to Reviewer 2 Comments
We thank Reviewer 2 for their insightful and helpful comments and the Editor for providing us with the opportunity to revise the paper. We have adapted the manuscript in line with the Reviewers’ comments. The responses to individual comments are described below in red and revisions made in the manuscript are also in red.
Point 1: - Pages 1-2. The introduction is too long and multithreaded; this section could be shortened.
Response 1: We have shortened the introduction as suggested. The Second paragraph of the introduction has been moved to the start of Section 4 and is highlighted in red (Lines 196-209). Other parts of the introduction that have been deleted are highlighted in red and strikethrough.
Point 2: - The Authors of this manuscript should present and discuss the ISPAD Clinical Practice Consensus Guidelines 2018 in the context of opportunities and limitations of low carbohydrate diet for youth with type 1 diabetes. Please include the references: Nutritional management in children and adolescents with diabetes. Pediatr Diabetes. 2018, 19 Suppl 27:136-154; and Exercise in children and adolescents with diabetes. Pediatric Diabetes. 2018, 19 Suppl 27:205-226. The authors refer to youth with type 1 diabetes in the introduction to the manuscript.
Response 2: Thank you for this suggestion, we agree that it is very important to cite the latest ISPAD guidelines. These are added to our section outlining the potential for risk of maturation deficits with low carbohydrate diets in children. We have added the following sentence to section 4.2 Lines 410-416 to refer readers to Pediatr Diabetes. 2018, 19 Suppl 27:136-154.
‘Readers are referred to the latest ISPAD Clinical Practice Consensus Guidelines on nutritional management in children and adolescents with diabetes for additional information on the importance of appropriate energy intake to achieve optimal body weight, growth and development [110]. The timing and amount of carbohydrate intake around exercise and physical activity presents additional challenges to the child and their family so must be carefully managed [111].’
We have also added a further potential negative of low carbohydrate diets in Table 3: ‘Possible maturational deficits in children’ which is highlighted in red.
An additional line has been added to section 3 on lines 135-136 citing the ISPAD guidelines:
‘These macronutrient guidelines are similar for children and adolescents with T1D [26].’
Point 3: - The individual sections in the manuscript, except for the introduction should be numbered (sections and subsections). Please add such numbering.
Response 3: This has been added and is highlighted in red. We have numbered the Introduction as section 1 as this is done in other Nutrients manuscripts
Point 4: - Page 2 (line 87): Please present the aim of this review in numbered points, in the order corresponding to the body of the manuscript.
Response 4: Thank you for your suggestion. This has been highlighted in red in the manuscript on page 2 line 83-90 so that it now reads as follows:
“The aims of this review are as follows: 1) to investigate the known pros and cons of LCDs in people with T1D; 2) to discuss whether there is potential for low and/or restricted carbohydrate diets in recreationally active individuals with T1D; 3) to discuss carbohydrate restriction strategies for elite or recreational athletes with T1D as a means to improve training adaptation alongside improved glycaemic control. It is hoped that this paper will highlight areas for future research so that patients, healthcare providers, and sports coaches can make informed, evidence-based decisions about whether LCDs are suitable, while understanding the potential risks and limitations.”
Point 5: - Page 2 (line 94): ‘History of carbohydrate restriction in people with type 1 diabetes’ – this section could also be shortened.
Response 5: We have removed a lot of the information describing the specific starvation diets used by Allen and Joslin. The parts that have been removed are highlighted in red with a strikethrough on page 3, lines 97-104. The references to the starvation diets have been kept in so readers can still find this information if they choose to.
Point 6: - Page 5 (line 189-223): ‘Low carbohydrate diets and type 2 diabetes’ – I suggest deleting the subsection since the aim of the manuscript as identified by the Authors refers to type 1 diabetes only.
Response 6: This section has been deleted.
Point 7: - Pages 7-9 (line 265-280): Please move Table 3 to Supplementary Table S1 (Supplementary material). Page 6 (line 249): Please refer to Supplementary Table S1 instead.
Response 7: Table 3 has been moved to supplementary material section. The numbers for tables 4-6 have been adjusted to reflect this and are highlighted in red.
Point 8: - Page 10 (line 347): ‘Potential risks of low carbohydrate diets in people with type 1 diabetes’ – reference to Table 4 is missing from this section.
Response 8: Thank you for highlighting this. We thought it might work better to add the following sentence to the end of Section 4 on lines 248-250 (highlighted in red) to avoid repeating the reference to this table and because the Potential Benefits and Potential Risks sections are under one section.
‘The following subsections will discuss the potential benefits and risks of a LCD in people with T1D and these factors are then summarised in Table 3.’
Point 9: - Page 14 – Please move Table 5 to page 12, above Figure 1.
Response 9: This has been done
Point 10: - Page 16 (line 523) – Error in reference number; In References (page 27) – incomplete information for reference no. 79.
Response 10: This has been added and is highlighted in red. Reference 79 has been amended and is now reference 34 (highlighted in red)
Point 11: - Page 23 - Table 6 - Further areas for research specific to athletes and carbohydrate intake in athletes with T1D – Please add in separate subsections: Conclusions and Future Directions.
Response 11: Thank you for your suggestions. In an earlier draft of the manuscript we had aimed to add a section within Table 5 titled “what we know about carbohydrate intake in athletes with T1D”. However, we decided to remove this because there are no studies that have explored carbohydrate intake strategies in athletes with T1D so all of our information comes from studies of athletes without T1D. This is why we chose to add in the table with future directions specifically for the athlete with type 1 diabetes to highlight the need for more research. In the text of section 6 Lines 3 556-558 we make the following statement to highlight that the studies cited in our manuscript are from athletes without T1D:
‘Adaptations to exercise under carbohydrate restricted conditions are yet to be tested in people with T1D, therefore the majority of this section is based on extrapolation of mechanisms that operate in people without T1D.’
Furthermore, Section 7 contains our overall conclusions and future directions so we did not want to duplicate this in a table.
Point 12: - Please attach a list of abbreviations.
Response 12: This has been added to page 1 and is highlighted in red
*Please note that the authors have added a few extra sentences or edited typos which are now highlighted in red throughout the manuscript.
Reviewer 3 Report
I want to thank to reviewers for this very interesting narrative review. The paper is really well written. Please see my comments as suggestions how the manuscript can be further improved. Some points might need further discussions, or at least a more sensible way to transfer them into studies or practice.
Please find below my comments:
· Abstract, first sentence: Please state if this is globally and additionally, we know that there is high regional variation in achieving glycemic targets.
· Abstract, sentence: “These liberal guidelines facilitate…”. I am not sure about that. We are still discussing that people with T1D should be encouraged to life a “normal life” like healthy individuals. This includes also flexible insulin regimen. As long as CHO to bolus insulin is matched correctly, and CHO overconsumption is not performed, a healthy lifestyle is still achievable including a good glycaemic control, with less hypoglycaemic episodes. This flexibility also includes days where people with T1D are consuming > 300 g CHO/day and other days where they are consuming for example less than 50 g CHO/day. A predefined maximum of CHO or kcal/day would be a massive decrease in flexibility.
· Sentence line 58: “This is complicated by the fact that subcutaneous…”. Please use also this review as evidence, since the intra-variability might be lower (for newer insulins) (https://www.sciencedirect.com/science/article/pii/S1557084308800103)
· “For more 106 information, interested readers are referred to Hill and Eckman [22] for numerous historical reports 107 of patients prescribed these starvation diets.” – I do not think that this sentence is needed.
· Line 120: “more sophisticated insulin delivery devices (insulin pens and pumps)”. Traditional insulin syringes were/are still accurate. Especially for small insulin doses (0.1 – 0.3 units) they might be more precise in dosing than pen/pump. In general, this sentence is difficult to understand. Please rewrite. Why should faster insulins or pump/pen be able via supporting systems to more precisely dose insulin? It is not more accurate; via supporting systems it might be that CHO to insulin is better matched (e.g.: via calculating IOB etc.).
· Line 123: Just for example, why it is important to match insulin to CHO and not CHO to insulin: We know from practice that people with T1D and obesity often inject a standardised dose of bolus insulin for meals, and then consume CHO based on circulating insulin. Due to the fixed amount of insulin they are not able to lose BW. However, the recommended way via DAFNE allows in an easier pattern to decrease CHO intake. I do not believe that this is a reason for obesity in T1D, it might be the other way around.
· Line 145: Please check the current literature (https://www.ncbi.nlm.nih.gov/pubmed/30362180). The main problem might be severe hypoglycaemia in the context of glucagon rescue injection. Regarding DKA: as long as basal dose is matching endogenous glucose production, there might be a low risk of DKA. If there is other medication used like SGLT-2 inhibitors, it might be that the risk of an “euglycaemic DKA” exists.
· Please do not use “patients” with T1D – based on a current consensus please use people/individuals
· Line 177: you are talking before of VLCD and then, that high CHO diets need to be investigated in T1D – please rearrange.
· I do not think that the T2D section is needed. I would recommend deleting.
· Line 283: Since we see similar trends in non-T1D individuals, I would not be going down the route of saying that this is a phenomenon in people with T1D
· Around line 302: A main trigger for weight gain might be the permanent glucose absorbing effect of the circulating basal insulin. If someone is managing the therapy by means of basal insulin to overcome partial CHO intake, then therapy should be optimised based on the relation of bolus-to-basal insulin. It might be that a partial cover of basal needs by higher doses of bolus insulin could lead to improvements in weight management – however, even during low CHO diet, this increases the risk of DKA and hypos.
· Line 317 – 330: Not so sure about that content. First, authors are saying that hypo treatment with CHO might be a further problem for weight management; then there is a jump from glucose vs. mixed meal consumption. It reads like the authors suggest consuming mixed meals for hypo treatment. I am sure that is not what the authors wanted to say. Please improve wording.
· 332 -333: Please take care with this sentence. This would mean that in all people with obesity insulin resistance is present.
· 340: detemir is not a NPH. What does long acting human insulin analogue mean?
· 345: Please take care – I think there is something mixed up. Insulin sensitivity is the amount needed of bolus insulin to cover a certain amount of CHO. This means if someone increased insulin sensitivity from “1 U bolus insulin covers 10 gr CHO” to “1 U bolus covers 20 gr CHO” = increase in insulin sensitivity. Additionally, if the total basal dose goes down = increased insulin sensitivity. Just by consuming less CHO = less insulin needed to cover, is not an increase in insulin sensitivity.
· Line 348: I think this sentence is already mentioned above.
· Line 354: iCGM (and in some regions rtCGM) is getting more and more funded by health insurances. I think it might make sense to discuss the VLCD and its effect on lowering HbA1c without increased hypoglycaemia due to CGM hypo alarms. Hypo-induced negative effects by VLCD can be avoided – hence beneficial effects of VLCD on micro-and macrovascular outcomes?!
· Line 360 – 372: I think this is already discussed in a previous section.
· Line 384: VLCD and micro-nutrients (multi-vitamin) supplementation as an option could be discussed here.
· Line 422: I am not sure about the part around social isolation. Different forms of diets (e.g. vegetarian, vegan, low carbing etc.) getting more and more common around Europe. I think that is in line with growing numbers of specific restaurants.
· Table 5, training status: If someone is better trained, it can be assumed that this person can stay longer (even for a relative intensity based on VT1/VT2) on lipid metabolism. This would mean (measured via indirect colorimetry) this person needs less CHO. However, the absolute workload would still be higher in comparison to an untrained workload. This would result in a higher CHO intake during exercise for the trained person in comparison to the untrained person. Please comment. (e.g. VT2 at 320 W vs. VT2 at 90 Watt).
· Line 565: I think the main problem of two or even more sessions per day in people with T1D might be that the bolus insulin will be dramatically reduced (and basal insulin). This means that on the one hand CHO are needed to refill hepatic glycogen stores (with exogenous insulin) on the other intra-muscular glycogen stores will be more or less continuously refilled due to GLUT-4 action (even under resting conditions between sessions). Please comment.
· Line 633 -639: I think that the main problem with the idea “exercising in a fasted state in people with T1D” is that there is always basal insulin (especially on MDI) circulating. Hence, lipid utilisation might be in general low. If basal insulin dose is dramatically reduced, then ketones will go up and be used as a “energy source”. Please comment.
Author Response
Response to Reviewer 3 Comments
I want to thank to reviewers for this very interesting narrative review. The paper is really well written. Please see my comments as suggestions how the manuscript can be further improved. Some points might need further discussions, or at least a more sensible way to transfer them into studies or practice.
We thank Reviewer 3 for their insightful and helpful comments and the Editor for providing us with the opportunity to revise the paper. We have adapted the manuscript in line with the Reviewers’ comments. The responses to individual comments are described below in red and revisions made in the manuscript are also in red.
Point 1: Abstract, first sentence: Please state if this is globally and additionally, we know that there is high regional variation in achieving glycemic targets.
Response 1: This has been addressed in the abstract and line 50 (highlighted in red) so that it says that this information refers to data in the United States
Point 2: · Abstract, sentence: “These liberal guidelines facilitate…”. I am not sure about that. We are still discussing that people with T1D should be encouraged to life a “normal life” like healthy individuals. This includes also flexible insulin regimen. As long as CHO to bolus insulin is matched correctly, and CHO overconsumption is not performed, a healthy lifestyle is still achievable including a good glycaemic control, with less hypoglycaemic episodes. This flexibility also includes days where people with T1D are consuming > 300 g CHO/day and other days where they are consuming for example less than 50 g CHO/day. A predefined maximum of CHO or kcal/day would be a massive decrease in flexibility.
Response 2: Thank you for your suggestion. We have adjusted this section of the abstract as recommended so that it reads as follows:
‘These liberal guidelines are designed to facilitate greater freedom around nutritional choices but they may lead to higher caloric intakes and potentially unhealthy eating patterns that are contributing to the high prevalence of obesity and the metabolic syndrome in people with T1D.’
Point 3: · Sentence line 58: “This is complicated by the fact that subcutaneous…”. Please use also this review as evidence, since the intra-variability might be lower (for newer insulins) (https://www.sciencedirect.com/science/article/pii/S1557084308800103)
Response 3: Thank you for your suggestion. This review has been added to the reference list and is highlighted in red. This section has been moved in accordance with Reviewer 2’s comments so that it is now in Section 4 (lines 196-209). This is highlighted in red.
Point 4: · “For more 106 information, interested readers are referred to Hill and Eckman [22] for numerous historical reports 107 of patients prescribed these starvation diets.” – I do not think that this sentence is needed.
Response 4: We have deleted a large portion of this section in accordance with Reviewer 2’s comments (lines 97-104). However, we feel that it is a good idea to keep the sentence referring readers to Hill and Eckman’s reports because they are likely to be of interest to readers.
Point 5: · Line 120: “more sophisticated insulin delivery devices (insulin pens and pumps)”. Traditional insulin syringes were/are still accurate. Especially for small insulin doses (0.1 – 0.3 units) they might be more precise in dosing than pen/pump. In general, this sentence is difficult to understand. Please rewrite. Why should faster insulins or pump/pen be able via supporting systems to more precisely dose insulin? It is not more accurate; via supporting systems it might be that CHO to insulin is better matched (e.g.: via calculating IOB etc.).
Response 5: We don’t say that new devices are more accurate, we do say they are more precise because they can give much smaller insulin doses with built in dose calculators which would allow greater ease around carbohydrate intake.
Point 6: · Line 123: Just for example, why it is important to match insulin to CHO and not CHO to insulin: We know from practice that people with T1D and obesity often inject a standardised dose of bolus insulin for meals, and then consume CHO based on circulating insulin. Due to the fixed amount of insulin they are not able to lose BW. However, the recommended way via DAFNE allows in an easier pattern to decrease CHO intake. I do not believe that this is a reason for obesity in T1D, it might be the other way around.
Response 6: Your point is well taken. As we know, some patients do take the same dose of insulin from day to day however this is not the point we are trying to make. We were trying to say that people with T1D are more likely to match their insulin needs to carbohydrate intake rather than carbohydrate intake to energy needs. This may ultimately be increasing obesity rates in this patient population.
Point 7: · Line 145: Please check the current literature (https://www.ncbi.nlm.nih.gov/pubmed/30362180). The main problem might be severe hypoglycaemia in the context of glucagon rescue injection. Regarding DKA: as long as basal dose is matching endogenous glucose production, there might be a low risk of DKA. If there is other medication used like SGLT-2 inhibitors, it might be that the risk of an “euglycaemic DKA” exists.
Response 7: Thank you for your suggestion. The paper by Seckold et al (2018) is referenced elsewhere in the manuscript. We agree that it is important to report the potential risk of DKA in individuals with T1D that are taking SGLT2 inhibitors so this has been added to lines 331-334 and is highlighted in red.
‘Of note, VLCD diets that have ketogenic effects may increase the risk of DKA in individuals taking sodium glucose co-transporter 2 (SGLT2) inhibitors since total daily insulin intake drops markedly [91] and there may be a risk of euglycaemic DKA [34].’
· Point 8 Please do not use “patients” with T1D – based on a current consensus please use people/individuals
Response 8: This has been addressed throughout the manuscript and has been highlighted in red
Point 9: · Line 177: you are talking before of VLCD and then, that high CHO diets need to be investigated in T1D – please rearrange.
Response 9: Thank you. We agree that this was an error so has been rearranged and is highlighted in red
· Point 10: I do not think that the T2D section is needed. I would recommend deleting.
Response 10: this section has been deleted
Point 11: · Line 283: Since we see similar trends in non-T1D individuals, I would not be going down the route of saying that this is a phenomenon in people with T1D
Response 11: Thank you for your suggestion. In this case, we would have to disagree with Reviewer 3. Although obesity is a universal problem, the rates of obesity and the metabolic syndrome in people with type 1 diabetes are extremely important to highlight and are a major reason that low carb diets may be advocated. This is important to highlight in this manuscript. We have added the following sentence to lines 284-286 to highlight this point:
‘A carbohydrate restricted diet may aid weight management in people with T1D, since caloric intake may drop and insulin dosage would likely decrease.’
Point 12: · Around line 302: A main trigger for weight gain might be the permanent glucose absorbing effect of the circulating basal insulin. If someone is managing the therapy by means of basal insulin to overcome partial CHO intake, then therapy should be optimised based on the relation of bolus-to-basal insulin. It might be that a partial cover of basal needs by higher doses of bolus insulin could lead to improvements in weight management – however, even during low CHO diet, this increases the risk of DKA and hypos.
Response 12: This is a very interesting point and we have seen that many patients, particularly on MDI, “over basal” and this leads to unnecessary snaking/caloric intake. However, we’re not too sure how to fit this argument in to this review as it seems somewhat tangential. We have edited lines 273-274 which is highlighted in red.
‘The rise in overweight and obesity may be related to intensive insulin therapy [58] coupled with a positive energy balance.’
Point 13: · Line 317 – 330: Not so sure about that content. First, authors are saying that hypo treatment with CHO might be a further problem for weight management; then there is a jump from glucose vs. mixed meal consumption. It reads like the authors suggest consuming mixed meals for hypo treatment. I am sure that is not what the authors wanted to say. Please improve wording.
Response 13: We have looked at this paragraph carefully and have made some minor edits. We certainly do not want to imply that hypoglycemia should be treated by a mixed meal. We were simply saying that frequent hypoglycemic events often leads to an increase in caloric intake that is not offset by reduced intake at other parts of the day.
Point 14: · 332 -333: Please take care with this sentence. This would mean that in all people with obesity insulin resistance is present.
Response 14: thank you for highlighting this. We have deleted the latter half of this sentence so that it now reads:
Line 305-306: ‘As outlined above, insulin resistance is common in people with T1D and the incidence is rising’
· Point 15 340: detemir is not a NPH. What does long acting human insulin analogue mean?
Response 15: thank you for highlighting this. We have changed the sentence so that it reads as follows on lines 312-314:
‘Chronic exogenous insulin use may be an important factor, as exposure to a long acting human basal insulin analogue such as insulin detemir or glargine
Point 16: · 345: Please take care – I think there is something mixed up. Insulin sensitivity is the amount needed of bolus insulin to cover a certain amount of CHO. This means if someone increased insulin sensitivity from “1 U bolus insulin covers 10 gr CHO” to “1 U bolus covers 20 gr CHO” = increase in insulin sensitivity. Additionally, if the total basal dose goes down = increased insulin sensitivity. Just by consuming less CHO = less insulin needed to cover, is not an increase in insulin sensitivity.
Response 16: we agree this was misleading so we have edited as follows on lines 318-322.
As LCDs have been associated with reduced insulin requirements in people with T1D, likely because of reduced carbohydrate intake [51,87,88], this reduction in insulin exposure might theoretically improve insulin sensitivity based on limited animal and observational studies [89]. Further studies are required that specifically examine the effects of LCDs on insulin sensitivity via insulin clamp methods and/or meal tolerance tests in people with T1D.
Point 17: · Line 348: I think this sentence is already mentioned above.
Response 17: Yes this point about claims that HbA1c is reduced with LCD was mentioned earlier but we wanted to include it here again because this is the main argument used by people that advocate these diets
Point 18: · Line 354: iCGM (and in some regions rtCGM) is getting more and more funded by health insurances. I think it might make sense to discuss the VLCD and its effect on lowering HbA1c without increased hypoglycaemia due to CGM hypo alarms. Hypo-induced negative effects by VLCD can be avoided – hence beneficial effects of VLCD on micro-and macrovascular outcomes?!
Response 18: This is a good point so we have added an extra sentence on line 333-335
‘However, the increasing use of continuous glucose monitoring with alerts to warn of impending hypoglycaemia may help to facilitate a LCD or VLCD [92,93].’
Point 19: · Line 360 – 372: I think this is already discussed in a previous section.
Response 19:
Point 20: · Line 384: VLCD and micro-nutrients (multi-vitamin) supplementation as an option could be discussed here.
Response 20: Thank you for your suggestion. The following sentence, highlighted in red, has been added to lines 379-381 in section 4.1.
‘Therefore, evaluation and monitoring of vitamin and mineral status in individuals with T1D following a LCD should be encouraged and multi-vitamin supplements may be recommended to reduce the risk of deficiencies.’
Point 21: · Line 422: I am not sure about the part around social isolation. Different forms of diets (e.g. vegetarian, vegan, low carbing etc.) getting more and more common around Europe. I think that is in line with growing numbers of specific restaurants.
Response 21: In this case we would like to keep this comment in. This section of the review is highlighting potential risks of LCD that may or may not apply to different individuals. We feel that highlighting the risks of eating disorders is very important in this manuscript.
Point 22: · Table 5, training status: If someone is better trained, it can be assumed that this person can stay longer (even for a relative intensity based on VT1/VT2) on lipid metabolism. This would mean (measured via indirect colorimetry) this person needs less CHO. However, the absolute workload would still be higher in comparison to an untrained workload. This would result in a higher CHO intake during exercise for the trained person in comparison to the untrained person. Please comment. (e.g. VT2 at 320 W vs. VT2 at 90 Watt).
Response 22: Following your comment we have decided to remove this point on training status from the Table. There is not much evidence to support that training status would have much of impact on glucose oxidation during exercise (J Appl Physiol (1985). 1997 Mar;82(3):835-40.; J Appl Physiol (1985). 1999 Oct;87(4):1413-20.)
Point 23: · Line 565: I think the main problem of two or even more sessions per day in people with T1D might be that the bolus insulin will be dramatically reduced (and basal insulin). This means that on the one hand CHO are needed to refill hepatic glycogen stores (with exogenous insulin) on the other intra-muscular glycogen stores will be more or less continuously refilled due to GLUT-4 action (even under resting conditions between sessions). Please comment.
Response 23: This is a great point and we simply don’t know the insulin dose requirements or the carbohydrate intake needs for individuals with T1D who are undertaking two or more sessions per day. We could speculate that with more bouts per day, much of the glucose disposal would be by insulin independent means. This highlights the need for more research in this area.
Point 24: · Line 633 -639: I think that the main problem with the idea “exercising in a fasted state in people with T1D” is that there is always basal insulin (especially on MDI) circulating. Hence, lipid utilisation might be in general low. If basal insulin dose is dramatically reduced, then ketones will go up and be used as a “energy source”. Please comment.
Response 24: This is also a good point. Surprisingly however, fat oxidation rates in well insulinised patients with T1D tend to be higher than non-diabetic controls (Phys Sportsmed. 2013 Nov;41(4):78-85. doi: 10.3810/psm.2013.11.2038; J Appl Physiol (1985). 2000 Apr;88(4):1239-46.).. Nonetheless, we agree that within an individual their own circulating insulin levels should influence the macronutrient oxidation ratio. Absolutely, if insulin is withheld for a prolonged period of time ketone production during exercise rises markedly (Diabetologia. 1977 Aug;13(4):355-65.)